# Workshops as Tools for Developing Collaborative Practice across Professional Social Worlds in Telemonitoring

**DOI:** 10.3390/ijerph18010181

**Published:** 2020-12-29

**Authors:** Niels Christian Mossfeldt Nickelsen, Roland Bal

**Affiliations:** 1School of Education, Aarhus University, 2400 NV Copenhagen, Denmark; 2Health Care Governance, Erasmus University, 3000 DR Rotterdam, The Netherlands; r.bal@eshpm.eur.nl

**Keywords:** social world analysis, collaborative practice, learning, telemonitoring, care ethics, workshops, joint action, COPD

## Abstract

Background: Lately, patients suffering from chronic obstructive pulmonary disease use telemonitoring services from home. We discuss three professional groups’ idea of good care in terms of living as a chronically ill patient. Methods: We scrutinize a workshop consisting of the following: (1) presentation of pre-workshop interviews focusing on good patient flows; (2) presentation of the participants’ photos illustrating their idea of the good life with telemonitoring; (3) discussion of what the three social worlds of care can do together. We understand workshops as learning events founded on the symbolic interactionist idea of learning as reflexism. That is, the process where participants make joint action an object of attention. Results: We propose that not only people, but also objects such as applications, gold standards, and financial arrangement are actively involved in hampering collaboration across social worlds. The contribution is a discussion of the contemporary challenges of technological intensification into healthcare processes seen as a learning event. Conclusion: Workshops constitute useful tools to understand more of how professional groups seek to adopt new technologies and learn about the larger structure of telemonitoring. Developing joint action among social worlds appears to be one of the main challenges of technologically driven innovation in healthcare.

## 1. Introduction

People live longer, and a growing number of people suffer from chronic illnesses. This justifies the use of healthcare technology at home. Increasingly, technology ingrains in people’s everyday lives and becomes co-constitutive for attempts to maintain a good life. It is this new and rapidly growing function for healthcare technology and the context it provides for healthcare professionals’ that we explore. The thrust of the article is reflections on how technologically-driven innovation is currently challenging professionals—and what this provokes. There is a growing body of evidence suggesting that healthcare services include technology on a completely different scale than before [1]. The target groups are much broader, and the endeavor is not only clinical and educational, but clearly also political and economic [2,3]. This complication relates to the question of the call to this special issue: How do we want to be cared for in the future? In light of this, we discuss the implications of healthcare technology for collaboration among different professional groups. We see this as a pressing challenge. Thus, our answer is: We want to be cared for by a collaborating healthcare system.

Healthcare increasingly embraces technologies introduced to ease, reorganize, or streamline a specific task; for instance, measurements and devices in telemonitoring [4,5,6]. When appropriate, such technologies constitute powerful tools for self-surveillance, education, and self-reliance [7,8]. Particularly in the Scandinavian countries, government agencies promote “welfare technologies” to reduce costs and improve quality [9,10]. Interestingly, the notion of welfare technologies allows for technologies that are not simply instrumental but embrace warm and caring relations between people and technology. Susskind and Susskind [11] have argued that telemonitoring has the potential to make expertise more accessible to patients. In order to lead to good practices across professional accountability systems, this however requires the dismantling of a number of resisting objects and barriers [12]. We take these barriers and the challenge to overcome them up here. Many different terms are used: telemedicine, telecare, and telehealth. These cover slightly different activities. We use telemonitoring and thus focus on the rehabilitation of patients suffering from chronic obstructive pulmonary disease (COPD) through feedback on measurements sent via an app to a healthcare worker (usually a nurse). Similarly, we consistently use “patient” rather than “client” or “citizen”. In doing so, we stress that the focus of the discussion is on people in need of help—and not just users or people with political rights and duties. COPD is a type of obstructive lung disease characterized by long-term breathing problems and poor airflow. The main symptoms include shortness of breath and cough with sputum production. COPD is a progressive disease, meaning it typically worsens over time. This often makes patients afraid of not being able to breathe—or even of dying. We analyze a workshop that we arranged to start a discussion on a collaboration focusing on the rehabilitation of COPD patients. We did this using analytical criteria such as, shared discursive spaces, shared commitment to action, and infrastructure. Thus, this study is not about patients but about professionals’ emerging collaborative practices in relation to all the COPD patients in this particular geographical region. We aim to answer the research question: How can an experimental workshop on COPD telemonitoring contribute to overcoming barriers between the social worlds of healthcare? We will return to the notion of social worlds in Section 2.3.

### Expert Recommendation: A Coherent Healthcare Sector?

In 2017, the National Board of Health in Denmark set up an expert panel to make recommendations for telemonitoring for COPD patients. Despite years of research, the report states there is a lack of knowledge about benefits from telemonitoring, what types of telemonitoring services work, and what preferences patients have. However, the report also notes, there are indications that telemonitoring leads to a better quality of life and that more research is needed to clarify these uncertain conditions. The purpose of the impending national rollout of COPD telemonitoring is to contribute to potential positive effects concerning both health and socio-economically. It is about making the individual COPD patient act on their own symptoms in order to achieve an increased quality of life, satisfaction, and security—and consequently fewer or shorter consultations, admissions, and re-admissions to hospitals [13].

The report provides detailed recommendations in terms of the target group, health professional content, responsibility, and collaboration. Among other things, the expert group proposes good examples of division of work between health centers, GPs, and hospital clinics. The recommendations indeed circle around spotting and dealing with inflammation at an early stage to avoid expensive hospitalization. COPD is clinically classified into categories A, B, C, and D (gold standards). This qualifies as evidence-based practice based on three parameters: spirometry, symptoms, and exacerbations [14]. According to this, telemonitoring services are relevant for patients in the D category (the most ill). The expert panel also states that GPs ought to refer to the municipal telemonitoring service after setting the patients’ individual alarm limits in collaboration with the health center nurses. At the health center, the telemonitoring nurses will then receive weekly measurements from the patient. The patients send measurements of oxygen saturation, heart rate, weight, and symptom score (answers to questions about the color and nature of saliva). After a few weeks, measurements ought to be evaluated in a collaboration between the GP and the health center nurse. The report emphasizes that telemonitoring also should be a service for the most ill patients while they are enrolled in an outpatient clinic.

The experts’ recommendations account optimistically for a coherent healthcare system deliberately collaborating across boundaries in the interest of patient and society. The report subscribes to neoliberal values (patients acting on their own symptoms) as well as to a cost reduction agenda (fewer re-admissions). Much points to the possibility that there is a gap between policymakers’ socio-technical imaginaries and rehabilitation in practice. This gap, we argue, calls for the scrutinization of barriers to support professional collaboration in technological development projects.

## 2. Learning across Professions

Studies in healthcare have shown that social interaction between professionals is crucial in enabling professionals to learn through their work and the problems they face [15,16]. Workshops as tools for learning about cross-professional collaboration is not something new. There have been many experiments, both in social work [17,18] and in healthcare [19,20,21]. We take up a contemporary problem of technological intensification of healthcare as a question of learning across social worlds. May et al. argue that four key barriers hamper sustainable telemonitoring services to patients with chronic diseases: (1) uncertainties about coherent service and business models; (2) lack of coordination across primary and secondary care; (3) lack of financial or other incentives to include telemonitoring within primary care services; (4) lack of a sense of continuity with previous service provision and self-care work undertaken by patients themselves [22]. Telemonitoring services may offer a cost-effective and safe form of care for patients with chronic illness. These four issues however lead to poor integration of policy and practice. Uneven integration likely stems from an incomplete understanding of the role of telemonitoring systems and subsequent deficient adaption to different healthcare contexts. It appears that research is lacking in this area leading to uncertainties about the best way to develop chronic disease management. Although professionals’ readiness to use telemonitoring [23] as well as facilitators and barriers in inter-professional collaboration have been studied [24,25]. We need to understand more of the barriers of professional collaboration across institutional settings. We scrutinize a workshop as an arena of learning, as a methodology for discussion of care values and for developing collaboration among health centers, GPs, and clinics.

### 2.1. Learning as Reflexism

Symbolic interactionism has its historical roots in the Chicago School of Sociology drawing on pragmatist philosophers like Mead [26] and Dewey [27]. Blumer [28] (p. 68) has made a seminal contribution to understanding joint action and learning. He argues that technologies (objects) are constituted through the meaning they have for those in relation to whom they are objects. That is, objects arise out of the way persons interact with them.

He proposes three theses for meaning-making: (1) people relate to their surroundings on the background of the meaning the outside world has to them; (2) they create this meaning in social interaction; (3) the social interaction is a continuous reinterpretation of meaning [28,29]. The implicated self/group is vis-à-vis the world instead of merely in the world [26] (p. 135). In collaboration, human agents piece action together by interpreting gestures, language, and objects. Hence, actors create joint actions by way of fitting their lines of action [28] (p. 70). According to symbolic interactionists, joint action across professional groups is an extension of conduct that emerges by way of constant mutual interpretation of ongoing activities and their meaning and effects. Mead called this “learning as reflexism”. This, we believe, is a useful theoretical background for our analysis of a workshop as a tool for developing collaborative practice in a complex technological society.

### 2.2. In-Action Ethics

The notion of in-action ethics is useful for elucidating emerging collaborative practices. A growing body of evidence is suggesting that practitioners are linking anticipatory ethics with their practice [30]. Healthcare professionals (and patients) are in other words struggling with ethical questions while they care. Instead of deciding in advance (and universally), what is “good”, people are challenged by situations. In other words, they act mindfully, and they improvise based on the stock of knowledge they bear [31,32]. The notion of in-action ethics draws among other things on the well-known educational idea of “the reflective practitioner”. Here, the practitioner, rather than from schooling, learns from the complexity of situations by interacting with the world and its demands and materiality [33]. The concept of in-action ethics involves ethical reflections in practice and opens the possibility that not only researchers identify ethics in their fields of study. In addition, practitioners learn about ethics while they carry out their tasks. The notion of in-action ethics is helpful in opening up the discussions at the workshop, and to understand more of the clashes, contestations, and meaning making among groups articulating different care values.

### 2.3. Social World Analysis

The core idea in social world analysis is that meanings of phenomena lie in their embeddedness in relationships. Universes of discourse [26] or social worlds are defined as shared discursive spaces [34]. Social worlds are universes of discourse and practice through which common symbols and activities emerge. Over time, social worlds may segment into multiple worlds and eventually merge with other worlds with which they share commitments [35] (p. 113). If the number of social worlds becomes large and conflictual, the whole is an arena. Thus, an arena is composed of multiple social worlds organized ecologically around issues of mutual matters of concern and commitment to action [35,36,37,38]. The social world framework is ecological, seeking to understand the nature of relations across groups of people and things in the arena. It is particularly attentive to situatedness and contingency. Since the 1980s, interactionist studies increasingly use the social world framework in regard to social and cultural aspects of technology. This encourages the exploration of virtual and technical infrastructures as deeply rooted aspects of social world analysis. The work of Star has greatly influenced thinking about technologies and infrastructures [39,40,41]. Together with Griesemer, she presented the notion of boundary objects arguing that sufficiently plastic standards and objects mediate relations among different groups of people and make it possible that they can work together without agreement [42] (p. 393). Fujimura also argued for the intertwined relationships between standards and interaction among multiple groups [43]. This is interesting in relation to chronic disease telemonitoring because figures sent from home (and other standards) mediate relations among healthcare professionals in new ways. Infrastructures are imbricated with the unique nature of each social world and may be seen as kinds of frozen discourses that form avenues between social worlds and into arenas i.e., larger structures [41,44]. Clarke suggests:

“to make a social worlds/arena map, one enters into the situation of values and tries to make sense of it starting with the question: What are the patterns of collective commitments and what are salient social worlds operating here”[35] (p. 113).

With additions and extensions from Foucault, Haraway, and Latour, Clarke developed social world analysis as a methodology that is useful in initial mapping and clarification of the studied field’s heterogeneous and complex character. Engaging in this epistemological hybrid, social world analysis provides an important instrument to stay with the relationality and ecology of the empirical site. This is helpful because it leads to questions like, what epistemic practices are enacted when professionals work with patients in various contexts [45]. This, we believe, is a good starting point for projects that focus on contested areas and emergent ethics. Thus, we embarked on this kind of analysis. We invited care providers from different social worlds as informants. Our informants are all involved (or expected to be involved) in telemonitoring. This fits well with Blumer’s idea that joint action takes place by mutually fitting lines of action. In that sense, we see the workshop as a social experiment for developing collaborative practice and understanding barriers in the telemonitoring arena [46].

In the following, we will extract a number of social worlds involved in the COPD telemonitoring arena. As part of this, we will seek to understand the nature of relations across groups of people, things, and actions taking place in this arena. The criteria for what constitutes social worlds in the analysis are: (1) shared discursive spaces; (2) shared commitment to action; (3) shared infrastructure that connects members of social worlds both internally and in relation to other social worlds.

## 3. Materials and Methods

The empirical focus of the analysis is the impending mandatory national rollout of COPD telemonitoring in Denmark as a collaboration among health centers, GPs, and clinics. This refers to the national financial agreement in 2016 and the digitalization strategy [47], as well as the government base from 2011. Additionally, it refers to guidelines developed by national and regional expert panels on telemonitoring that we discussed earlier [13,48,49]. However, due to security issues in relation to the common ICT platform, the government has postponed the rollout several times; allegedly, they will fully implement it in 2021. After years of pilot projects, the Danish government has decided to implement COPD telemonitoring in all municipalities. Many telemonitoring projects fail because of a lack of care integration. There have been problems with integrating telemonitoring in the healthcare system for many years [22]. The Danish case is interesting because it will change that. Thus, this is a unique and problematic initiative because so many interests are at stake. This deserves research attention. The plan begs the question, how will the participating professional groups collaborate across social worlds?

Before the workshop, we individually interviewed all participants. We used a semi-structured model to reach out for the informants’ beliefs about COPD telemonitoring and good patient flow. A research assistant transcribed all interviews verbatim. We analyzed the interviews before the workshop. The analysis focused on articulated care values and the informants’ ideas concerning good collaboration across social worlds of care. In addition, before the workshop, we instructed the participants to take three pictures of what makes a good life with COPD telemonitoring (see Figure 1, Figure 2, Figure 3 and Figure 4). As part of this, we instructed them to take notes in regard to each picture in a logbook [50]. There were 16 participants at the workshop:Four COPD patients involved in telemonitoringFour GPsTwo health professionals from the outpatient clinic (specialist physician and managing nurse)Six nurses from the municipal health center

We had the workshop at a health center and it took three hours. The first author opened the workshop by presenting central points articulated by the health center, the GPs, and the clinic (see Scheme 1). Following, one by one, the participants presented their pictures and notes in relation to their beliefs of what makes a good life with telemonitoring from home. Then followed a discussion of how to collaborate on telemonitoring services and a discussion of barriers. The discussion focused on barriers in terms of different infrastructure. A research assistant made a detailed summary of the workshop. Through this method, we wanted to: (1) mature the participants’ views by making it clear in advance what the workshop was all about; (2) further clarification of the participants’ views by way of several contacts over a timeline; (3) create engagement around the forthcoming collaboration. Thus, the idea in the two-step analysis was to prepare, expand, and nuance data. In the Results section, we analyze the social worlds of care that we identified by way of interviews and discussions at the workshop. As mentioned, we use the following criteria: (1) shared discursive spaces; (2) shared commitment to action; (3) shared infrastructure.

## 4. Results

### 4.1. Three Social Worlds of Care

In the Results section, we analyze the social worlds of care we have identified by way of interviews and discussions at the workshop by way of the following criteria: (1) shared discursive spaces; (2) shared commitment to action; (3) shared infrastructure.

### 4.2. Social World 1. Health Centre Telemonitoring as Continuing Contact Based on Weekly Home Measurement

At the Telemedicine Center (TMC), located at the municipal health center, the telemonitoring service grounds on “Bring Your Own Device” (BYOD). Patients may download an app on their own tablet or smartphone. At home, they measure oxygen saturation and heart rate with a small appliance (see Figure 1 presented by a nurse at the workshop). Together with their symptom score, through the Open Tele application, the patients send figures to the health center. They abide challenges in relation to data legislation by using the patients’ own devices. Data transfers from the patients’ devices to the health center are done solely in a secure encrypted system. Thus, in accordance with the EU Personal Data Regulation (GDPR), no data transfers to the GP or the hospital. However, if the patient wants, by bringing their own device, they can share data with the GP and the clinic.

The patient-interface instructs primarily on how to carry out, understand, and send figures. In addition, it provides general educative information about COPD (see Figure 2 and Figure 3). The nurse-interface comprises a bell system and some graphs. If a green bell appears on the nurse’s screen, everything is fine. If a blue bell appears, the patient has forgotten to submit measurements. If a yellow or red bell appears, there is a mismatch between current and former measurements. A red bell indicates an impaired condition. If a red bell shows, the nurse immediately needs to call the patient. The patient may for instance have reported cough changes or changed color of mucus. The nurse and the patient can then keep track of changes on a graph. By way of telephone calls and text messages, they communicate on the patient’s condition and the nurse offers self-care advice.

In opposition to the GPs, the nurses have time to talk with the patient. While the GPs have many patients, the health center nurses monitor a relatively small amount of patients. All the nurses we have interviewed explain they find this set-up relevant, ethical, and safe. One of the biggest advantages is that the telemonitoring service ensures continuous contact with the patient. Our patient informants praise that they have fast and continued contact with the same nurse. They also explain they like to be subject to monitoring and knowing that “you always have an empathetic conversation partner nearby” (interview with patient). To them, having continual contact with a well-known nurse is indeed valuable, they explain.

Formally, the health center’s inclusion criteria are narrow (gold standard D); however, in practice, they are broad and pragmatic (gold standard A, B, C, D) [14]. A health center nurse explains at the workshop,

“We do not worry about how good or bad the patients are in terms of categories A-D. If they think there is a need for telemonitoring, we just include them in the program”.

That is, the telemonitoring service is open and welcoming. However, in order to take part in telemonitoring, the patient needs to have a formal diagnosis (COPD) and to live in the municipality [49]. The unique feature of this telemonitoring model is the division of patients into two groups. The self-monitoring patients simply download the app and start monitoring themselves. The assumption is that they may learn from systematically collecting data about their lung function without advice from health professionals. The second group receive feedback from nurses every week. Typically, the patients in this group send figures to the health center one to two times a week. Subject to an agreement with the health center, they can shift between the two ways of participating.

According to nurses and managers, the municipality is in the lead in Denmark regarding telemonitoring for chronically ill elderly. This is among other things because the Municipal Director of Health and Care is actively involved in both national and regional boards. He also often acts as a health technology informant in the media. A nurse states at the workshop,

“We have a director who very much wants us to be the telemonitoring vanguard. This is something that requires that you stand firm—and it costs. There are some costs when you really want to try something new”.

This sense of pioneering relates interestingly with the issue of telemonitoring as a business case. According to the Danish organization of the healthcare system (The DRG), the municipality is required to pay up to one-third of the expenses for an admission to the hospital. By detecting inflammation at the earliest possible time, the municipality may not only deliver high-quality service, they may also save money. Thus, the health center social world comprises the shared discourse of pioneering, the commitment to action based on feedback, and the infrastructure of the bell-system. However, according to the director, telemonitoring still does not constitute a convincing business case.

### 4.3. Social World 2. COPD Care as Gold Standards Practiced by GPs

There are too few GPs in southwestern Denmark. Hence, each GP has a portfolio of around 1700 patients. They are busy. This became evident for the first author when he opted for access to GPs as informants. In fact, he was in several cases not able to call a GP without identifying himself as a patient with identification numbers (CPR). GPs e-mail addresses are often not publicly accessible. When he showed up in person at the clinic, in several cases, he talked to a secretary only because the GPs had waiting rooms full of patients.

Nevertheless, we succeeded in having five 30-min interviews with GPs before the workshop. All our GP informants follow the accredited DSAM (Danish Association of General Practitioners) stratification of gold standard A, B, C, and D. The gold standards are set-up in a chart demonstrating what characterizes A, B, C, and D in terms of symptoms and exacerbations (Figure 4). In addition, the Figure shows prescribed medication under certain circumstances, vaccination, advice on movement, smoking cessation, etc. The chart is practically encased in plastic so you can read both front and back, and bring it along—a practical, resilient, and easily accessible tool for GPs.

Moreover, the gold standard has another important function. It divides work between GPs and the clinic. While the GP’s take care of gold standards A, B, and C, the clinic is responsible for gold standard D. A, B, and C patients are offered one or two yearly lung function checks at the GP, in addition, they may visit the GP as often they like.

All the GP services reimburse in relation to a detailed tariff agreement concluded between the ministry of health and the GPs’ union (PLO). These tariffs are negotiated based on very specific operations such as a vaccination, a 5-min phone call, a 10-min consultation, etc. In addition, the GPs are assigned a number of mandatory conditions. For instance, a so-called chronic grant. That is, GPs receive a one-time annual benefit that covers for chronically ill patients with diabetes, dementia, and COPD. This benefit covers all care; no matter how much or how little they do for the patient. According to our informants, this is far from an incentive for GPs to engage in chronic patients (which is exactly what telemonitoring requires). In contrast, it makes the GPs withdraw in relation to chronically ill patients. GPs also have to implement individual COPD action plans. COPD action plans and COPD telemonitoring are both tasks coming from above targeted at the same patients, but without clear linking between them. It is unclear to all our informants how they support each other. The GPs, not surprisingly, find it obfuscating that such decisions are at the same time mandatory and uncoordinated.

The GPs explain they fear that telemonitoring disrupts an accredited practice that works well (gold standards). However, during the interviews and workshop discussions, it became clear that COPD patients are diverse and indeed have different needs. While some are in employment, others are very ill. Others, again, hide and are embarrassed. A complication is that COPD patients often see the disease as self-inflicted because they have not been able to stop smoking. Concerning patients coming to the GP with their telemonitoring figures on a tablet or phone, the GPs explain that they do not have the time to assess their implications seriously. This is because the standard consultation time is estimated at 10 min and thus there will be no time for reading an app that the GP does not immediately recognize.

Summing up, what are the GPs shared commitment to action in relation to COPD rehabilitation? First, the GPs find the infrastructure of gold standards useful as instructions about what to do with COPD patients with various needs. Second, compared to the fact that the GPs find themselves in privately owned businesses, being subject to orders from above frustrates them. Thus, they share a discursive space focusing on insensible working conditions. For instance, they agree that there is no financial incentive or other to engage in COPD-telemonitoring. The one-time annual chronic benefit, in particular, draws on their commitment.

### 4.4. Social World 3. COPD Care as Self-Help Treatment Practiced by the Clinic

In the pre-workshop interview with the specialist physician at the clinic, he explains,

“The literature shows us that at present, there is no safe knowledge about which patients benefit from telemedicine, what types of telemedicine services work and what preferences the patients may have. However, a number of studies show significant improvement in quality of life in the telemedicine supported treatment group compared to the control group. We cannot rule telemedicine treatment out, but there is a need of more research”.

The doctor believes telemonitoring may be an important safety factor for vulnerable COPD patients. However, there is a lack of documentation that telemonitoring does in fact reduce the risk of re-admission to hospital. He explains, “we are afraid that telemonitoring ends up as a waste of the patients’ time”. Although the national rollout plan and guidelines propose that clinics ought to monitor gold standard D patients [13], the clinic has not yet started this. However, as an alternative, the clinic has launched a number of collaboration projects with municipalities about so-called self-help treatment. As part of self-help treatment, the doctor prescribes antibiotics and prednisolone in advance. As such, it is already in the patients’ homes in case of an inflammation. Since the clue is that the medication is already in the home, helped by a nurse, the patients are able to start self-treatment immediately. According to the specialist physician, it is a matter of getting the treatment started as quickly as possible after they have identified the first signs of inflammation. According to him, the core question is, is telemonitoring or self-help treatment after all the quickest way to prevent further inflammation? The group of doctors at the clinic, he explains, conceive telemonitoring as a monstrous set-up in relation to the simple aim of initiating treatment rapidly. According to the specialist physician, we need to do what we can to avoid re-admission of patients to the hospital.

At the workshop, the managing nurse at the clinic presents her pictures and notes. Her photos are all screenshots of referrals. Thus, she elucidates that the clinic engages in many kinds of cross-site collaboration. Moreover, she emphasizes that it is a priority for the clinic to buy devices for the most ill COPD-patients. The clinic is committed to self-help treatment—not telemonitoring. They share a discursive space focusing on the fastest way to identify inflammation and starting treatment, arguing that telemonitoring is a monstrous set-up. They prioritize the quest for identifying inflammation at an early stage by the infrastructure of prescribing medicines in advance—and thus having them ready in the home.

### 4.5. Summing Up

The health center takes the lead in the telemonitoring service and invites the other social worlds to participate. The municipal nurses have time allotted to monitor figures and give feedback. Undoubtedly, telemonitoring is a municipal strategy and investment. The health center has developed the necessary applications and procedures for a well-functioning rehabilitation based on telemonitoring. As we have argued, this has to do with a shared discursive space in terms of a positive attitude towards technology, but also with the fact that the municipality must pay a third of the costs of hospital admissions. In terms of this, the municipality is betting on preventive telemonitoring services to offer quality care and save money. The municipal managers and nurses share a discursive space and commitment to action based on the Open Tele app and the bell-system infrastructure. Unfortunately, to make telemonitoring a good business case seems to have long prospects. We will return to this in the discussion.

The GPs share a commitment to action based on the infrastructure of gold standards. All our GP-informants follow the accredited gold standards prepared by the Danish Association for General Practitioners. This implies that GPs take care of gold standard A, B, and C patients. However, the GPs have many patients and little time. Due to negotiations with the GPs union, the GPs are now not only required to take part in telemonitoring, in addition, they have to develop and monitor yearly COPD treatment plans. Finally, yet importantly, they are subjects to a chronic grant, a lump sum payment that is independent of what they actually do for chronic patients. These reforms press the GPs to perform certain centrally decided interventions without any incentive; consequently, the GPs share a discursive space of telemonitoring being a waste of time.

The lung medical clinic provides a shared discursive space and commitment to action based on the infrastructure of self-help treatment in relation to a number of nearby municipalities. At the clinic, the government’s mandatory plan to rollout COPD telemonitoring nationally does not seem to have any significance whatsoever. As mentioned, the clinic takes care of gold standard D patients. The specialist physician in charge raises serious doubts about whether telemonitoring is adequate at all in relation to these patients. Instead, the clinic promotes self-help treatment plans in collaboration with nearby municipalities. The point is to initiate treatment quickly in case of a crisis by way of storing medicines in the homes. In addition, the clinic focuses on purchasing expensive appliances for the most ill patients.

## 5. Discussion

The Danish Government hopes to see a healthcare system coherently collaborating across professional boundaries in order to implement COPD telemonitoring as a mandatory operation in all municipalities as soon as the ICD platform is available in 2021. Unfortunately, coherent collaboration is not what we see. Different social worlds of discourse and commitment to action struggle with practical questions tied to their local performance regimes. What we have observed are rather three social worlds that act in accordance with local values, priorities, and incentives [31,32]. That said, the identified social worlds also reach out to each other. They reach out to create joint action across sites and as they do that, they face a number of barriers in terms of different infrastructure such as financial arrangements, admission systems, and care values. In collaboration, seen with Blumer, they try to piece action together across sites at the workshop. That is, systematic collaboration across social worlds comes to be as a mutual interpretation of activities, objects, and effects. This fits well with what Mead called “learning as reflexism” [28] (p. 70). In accordance with this, the interviews, observations, pictures, logbooks, and discussions at the workshop displayed the distinctive features of three social worlds and clarified that the emergent shared commitment to action rests on features shaped by the particularities of each social world [20].

The discussions at the workshop clarified that the genuinely existing commitment to mutual action in the interest of the patient is undermined by particularities such as care values, belief systems, and financial arrangements (Reich, Rooney and Hopwood, 2017). At the workshop, however, the participants (and social worlds) reached out. They listened carefully to each other and opened up. By way of showing photos and explaining why exactly they brought this photo as an illustration of good cross-world COPD rehabilitation, they creatively shared their commitments. Thus, we witnessed that they were actually able to center the question; what is a good process across social worlds from the perspective of the patient? It appeared to us that they became aware of what is going on in the larger scene. The fact that policymakers have a strong voice attached to the telemonitoring arena is not something nurses necessarily realize in a busy day. As such, they became aware of the potentials and barriers of cross-social world collaboration. This is not to say that nurses and doctors are ignorant, rather that they now could see telemonitoring as part of the larger structure.

The workshop ended with a focused discussion on a common commitment to action. What could they do together? A nurse proposed that GPs and health center nurses could set alarm limits together as a fixed procedure before telemonitoring started. A GP announced that patients would benefit if GPs and nurses shared measurements systematically along the way and evaluated the process together. Could GPs and nurses have telephone meetings every three months? The participants talked about principles for keeping each other informed about patients as they transgress between the health center, the GPs, and the clinic. However modest, we believe it is an achievement that the workshop created this kind of initiative to overcome barriers. After the workshop, the health center nurses initiated three meetings with GPs to define and implement new procedures on these topics.

The learning that seemed to appear from the workshop was a matter of fitting lines of action as the mentioned expert panel emphasizes [13,49]. The national expert panel exactly proposes that GPs should be actively involved in setting alarm limits and in evaluating their patients’ involvement during telemonitoring. The expert panel calls for ambitious and demanding collaborative practices. In spite of all the barriers, this is exactly what the healthcare professionals now strive to realize.

Thus, much seems to indicate that the workshop managed to open up the social worlds towards each other. Before the workshop, the GPs and the clinic professionals had hardly heard of the telemonitoring services. In addition, it was new to the health center that the GPs receive a chronic grant that effectively limits their effort. Finally, yet importantly, it came as a surprise that the clinic does not appreciate telemonitoring at all and even promotes self-help treatment as an alternative. In accordance with what we are lining up here, we propose, the workshop occasioned learning as reflexism in the sense that the participants’ exchanges of care values and infrastructure helped to get a more complex understanding of the arena of telemonitoring and chronic illness management. We foresee, however, that local performance regimes continue to constitute barriers to create more joint action across sites. For instance, at the workshop the GPs declared they do not have the time to learn about the telemonitoring app. We will sum up the discussion by proposing that the participants learned to cope with telemonitoring as a contested arena inflicted with many values, ambitions, and practices. However modest, this result, we believe, is an achievement.

### Changing Policy Contexts

When taking a closer look at the arena of COPD telemonitoring, there are more social worlds than those we have so far identified. The government does not just herald a new mandatory practice in this area. Currently, it is also contested how all this is to be organized. At a meeting, the Director of Health and Care says,

“Telemonitoring hardly constitutes a convincing business case now. On the political arena, nobody seems to be interested in telemonitoring. Whereas the former Liberal government promoted welfare technology, the new Social Democratic government wants nurses and warm hands in all places—not technology. Rather, currently the discussion is, from what organizational entity the citizens should get their teleservice. Perhaps, in the future there will be a few telemonitoring centers spread out over the country—perhaps one center in each region”.

This indicates that not only the national rollout but also the strategy of quality and savings at a municipal level stands on shaky grounds. Since telemonitoring does not make a convincing business case yet, now it is uncertain what will happen to the former government’s plan for a national rollout of COPD telemonitoring. Telemonitoring may after all be too expensive to uphold in a single municipality. It appears there are simply not enough COPD patients to finance a well-functioning system locally. Therefore, negotiations are going on among the Ministry of Health, the regions, and the municipalities about where to locate possible new and larger telemonitoring centers. The government may after all lift telemonitoring out of the single municipality. Consequently, we may face a battle over where a number of regional telemonitoring centers and the relevant human competencies are to be located. The Director of Health and Care does not hide his willingness to attract such a center and thus be able to serve nearby municipalities. In order to be able to offer a model that integrates primary and secondary care and to attract a possible new center, the health center currently prepares telemonitoring systems not only for COPD patients, but also for diabetes, heart failure, and for cancer patients. There is some evidence that if you can scale up the number of diagnoses and thus the chronically ill, then you can also develop a sustainable model. Much seems to indicate we are facing a political battle about how to organize telemonitoring services for chronic patients in Denmark.

## 6. Conclusions

Collaboration among social worlds of care appears to be one of the main challenges of technologically driven innovation in healthcare now. We want to be cared for by a healthcare system that is able to collaborate across sites and offer sustainable and integrated healthcare processes. New healthcare technologies make visible how difficult this is. Drawing on symbolic interactionism and social world analysis, we scrutinized the implementation of telemonitoring services as a matter of developing collaborative practices among a municipal health center, GPs, and an outpatient clinic. Former research has pointed to uncertainties with regard to the business model as well as uncertainties in relation to coordination among the primary and secondary healthcare sector. We discuss the question of adaptation of telemonitoring services as a collective challenge among different social worlds of care characterized by shared discursive spaces, shared commitment to action, and shared infrastructure. The empirical material comes from preparing and holding a workshop. On that ground, we discuss the research question: How can an experimental workshop on COPD telemonitoring contribute to overcoming barriers between social worlds of healthcare? The notions of in-action ethics, joint action, learning as reflexism, and infrastructure were helpful in the identification and analysis of interaction among three social worlds. These notions supported scrutinizing what kinds of standards, clinical targets, and financial aims that each social world seeks to realize. From this, it appeared that the emergent and fragile collaboration across social worlds rested on the particularities of the accountability systems of the identified social worlds. The workshop was set-up as an experiment for discussions of care ethics and for developing collaboration across social worlds in relation to the telemonitoring of COPD patients. The aim was to analyze emergent professional collaborative practice. The interviews, observations, photos, logbooks, presentations, and discussions at the workshop not only elucidated social worlds characterized by different care values and practices; in addition, it illustrated that different financial incentives hamper further integration of primary and secondary care practices. That is, different ideas of what is good care constitute barriers to further development of cross-site collaborative practices. Finally, yet importantly, the workshop was an occasion for learning where participants’ exchange of values and infrastructure gave access to the larger structure of COPD telemonitoring. It appears however that even if the participants in the future should communicate more and offer more mutual feedback (as they say they will), local performance regimes will still constitute obstacles to create cross-site collaborative practices. We propose however that the workshop participants have learned to cope with telemonitoring as a complex and contested arena inflicted with many different values, practices, intersecting admission systems, and financial arrangements.

## Data Availability

We refer to the Data Availability Statements in MDPI Research Data Policies (https://www.mdpi.com/ethics).

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
