# Peer review of "Workshops as Tools for Developing Collaborative Practice across Professional Social Worlds in Telemonitoring"

_ijerph, 2020, doi:10.3390/ijerph18010181_

Round 1

Reviewer 1 Report

Telemonitoring for patient with COPD is an important challenge in medical field. However, Please describe whether the subjects selected in this study are representative of all COPD patient. Also, telemonitoring is not a recently developed method. It has been tried and applied in many other countries. Please add why Denmark should be emphasized on telemonitoring for COPD patients compared to other countries. Please insert the process and contents of WORKSHOP with flowchart.

Reviewer 2 Report

Thank you very much for giving me the opportunity to review the manuscript entitled "Workshops as tools for developing collaborative practices across professional social worlds in COPD telemonitoring".

The main concern about this paper is that the references throughout the paper should be reviewed. Citation of references within an abstract is inappropriate. Also, despite the fact the authors can include references in the methods and conclusion sections, but it is advisable not to include any. The citations and references of the manuscript are old and outdated.

Reviewer 3 Report

I have reviewed the paper,

Manuscript ID:  1001738

IJERPH is a high-level research journal.

Although I consider that this article has a low level of scientific rigor, I do consider that the subject is important.

For this reason, I allow myself some suggestions in case they could be useful to the authors for future publications.

Acronyms should not appear in the title, nor in the abstract

the abstract does not identify the methodology. What the authors describe is not methodology

it is not identified from the beginning what the acronym COPD means

In keywords, social network analysis appears, and this method does not appear in the paper.

I am an expert in methodology, and I have not seen the method described, nor the theoretical part that could be applied to the paper, nor the variables of the method that this research measures.

The introduction is too short.

It does not address the key parts that an introduction and justification must have.

The authors do focus on the subject, but they do not describe the part of networks that they should do in this section as a theoretical context.

The objectives are not clearly described.

Nor is the impact or why this research is clear.

In the material and method section, the authors describe theoretical aspects that should not appear in this section.

Part of this content should be in the introduction.

In the material and method section, the authors must clearly describe how the research is carried out.

That is, for each objective, what variables will be used, the definition and how they will be measured.

also the tools to collect the data, the type of analysis, etc.

In the results and the discussion, the authors did not structure the contents adequately.

That is, at the time of reading the paper, the reader is not able to identify which result responds to which objective, or which result is being discussed.

References are not up to date

Obviously, it is necessary to cite the initial sources, but the authors must carry out an updated review on the subject, especially in the last 5 years.

Once again, I do not see specialized literature on the Social Network Analysis methodology

Reviewer 4 Report

The article is interesting. I think the introduction and methodology could be more clearly presented.The introduction was a bit general Could there be a clear RQ?  COuld there be more information about the interviews and the idea of a two step process, with interviews first and then a workshop.  provide an argumentation for this process. 

I needed to google COPD, so there should be a bit more explaining about these patients and why they were selected. 

Round 2

Reviewer 3 Report

I have carried out the second revision of this paper

The authors have improved the content

I think the following tables and photos can be improved.

Also, authors should review how they have cited the references.

Improve table 1

Improve photo 4

Improve references
